# A Software Defined Radio Based Anti-UAV Mobile System with Jamming and Spoofing Capabilities

**DOI:** 10.3390/s22041487

**Published:** 2022-02-15

**Authors:** Renato Ferreira, João Gaspar, Pedro Sebastião, Nuno Souto

**Affiliations:** 1Department of Information Science and Technology, ISCTE-Instituto Universitário de Lisboa, 1649-026 Lisboa, Portugal; joao_filipe_gaspar@iscte-iul.pt (J.G.); pedro.sebastiao@iscte-iul.pt (P.S.); nuno.souto@iscte-iul.pt (N.S.); 2Instituto de Telecomunicações, 1049-001 Lisboa, Portugal

**Keywords:** jamming, GPS spoofing, radionavigation, software defined radio, unmanned aerial vehicles

## Abstract

The number of incidents between unmanned aerial vehicles (UAVs) and aircrafts at airports and airfields has been increasing over the last years. To address the problem, in this paper we describe a portable system capable of protecting areas against unauthorized UAVs, which is based on the use of low-cost SDR (software defined radio) platforms. The proposed anti-UAV system supports target localization and integrates effective jamming techniques with the generation of global positioning system (GPS) spoofing signals aimed at the drone. Real-life tests of the implemented prototype have shown that the proposed approach is capable of stopping the reliable reception of radionavigation signals and can also divert or even take control of unauthorized UAVs, whose flight path depends on the information obtained by the GPS system.

## 1. Introduction

Although drone legislation [1] exists, many unmanned aerial vehicle (UAV) operators choose not to comply with it, generating various types of incidents, either voluntarily or involuntarily.

Several solutions have been presented to try to solve this problem, such as birds of prey, capture nets, or even jammers as is the case below. For example, the Dutch police have trained some eagles and hawks, so that they can chase and hunt UAVs whenever necessary [2]. In Russia, a non-destructive solution was presented, the Rex-1, which comprises an emitter of an interference signal against the global navigation satellite system GNSS signal [3].

Several studies on detection and jamming of these devices have also been carried out in order to combat this problem. The paper [4] emphasizes the guidelines for drone successful jamming while presenting some solutions implementation for drones malicious function prevention and drones role in jamming efficiency improvement. In [5], the authors present commercial-of-the-shelf (COTS) applications components for various military purposes. Solutions aimed at military application based on software defined radio (SDR) systems are considered, as well as a multifunctional system development base for detection and identification of malicious drones. Implementation of RF surveillance based on COTS components is described, which resorts to ADALMPLUTO and USRP platforms. An SDR is a radio communication system that contains various reconfigurable software-based components for processing and converting digital signals. Unlike traditional radio communication systems, these radio devices are highly flexible, versatile and configurable. In this paper, the SDR used was the BladeRF x40 from the manufacturer Nuand. In past studies, the present authors have also studied techniques to solve the problem. In [6], we evaluated the use of low cost SDR platforms for the implementation of a jammer able to generate an effective interfering signal aimed at the GPS navigation system. Using a programmable BladeRF x40 platform from Nuand and the GNU radio software development toolkit, several interference techniques were studied and evaluated, considering the spectral efficiency, energy efficiency and complexity.

The different types of UAVs, the regulatory laws for UAV activities, their use cases, recreational and military UAV incidents have already been discussed in [7]. Various techniques for monitoring and preventing UAV attacks were studied and described along with case studies in [8]. In [9], the authors conducted extensive market research of 581 counter-unmanned aerial system (C-UAS) systems from 282 manufacturers in 39 countries around the world based on the literature and practice, making it a four-way statistical analysis. Based on this, they classified and summarized the most advanced weapon systems according to different technologies. In [10], the authors presented the results from the field test of UAV GPS jamming by using software defined radio (SDR) and GNU radio software. The field test was carried out to study the performance of the jamming signal produced and the effect of elevation angle on the jamming range. The test subject used in this field test was a DJI Phantom 4 Pro, a commercial consumer grade drone.

A different approach to jamming consists in gaining control of the UAV by spoofing the control commands, which normally rely on radio waves. However, different UAVs may operate with different communication protocols, depending on the UAV brands or models. Furthermore, since some UAVs may communicate under a dedicated communication protocol, this command-and-control spoofing solution may not work. It becomes simpler to perform satellite navigation signal spoofing, since global positioning system (GPS) signals are widely used by most UAVs, especially for autonomous operations [11].

Studies have already been carried out with the subject of spoofing the GNSS signals against systems that depend on these, namely UAVs and smart grid devices. It was demonstrated that a civil UAV could be “steered” by a spoofer by moving its perceived location in the opposite direction of the desired motion. Coarse, short-term control of the UAV was demonstrated in all directions (east, north, and up) during the tests [12]. There have been a few other studies and investigations in the area of GPS spoofing against UAVs. In [11,13], the authors studied and exploited the vulnerabilities of GPS systems in drones in order to hijack or gain control over the aircrafts. Another example of exploiting GPS vulnerability is that of the UnicornTeam, a group of security researchers focusing on systems based on radio technologies. They proved, in [14], through several approaches, that it is possible to spoof a GPS receiver, which the present authors have also demonstrated in [15].

Some more robust systems have already been developed in order to solve the problem of unauthorized drones. For example, the system presented in [16] is based on the identification of communication protocols for later disruption and/or replication and is designed for implementation in a distributed system. While it may be very effective against a few specific UAVs, the implementation complexity can be very high and the portability may be limited. The systems proposed in [17,18] also constitute anti-UAV approaches but include only the blocking solution, i.e., they do not have the ability to gain control over the invading drone and force it to land in a safe zone.

It is important to note that other, very different, approaches exist. Such is the case of the system presented in [19], which uses a more basic technique, i.e., resorting to a net to capture the invading drone. The system presented in this article uses non-destructive methods, using radio frequency.

Motivated by the work above, in this paper, we study a system capable of neutralizing unauthorized UAV flights using low cost SDR platforms. The implemented system is capable of jamming the target UAV while also transmitting false GPS signals so as to block communications, redirect or even gain control of the vehicle flying over protected areas. For evaluating the behavior of the system’s operation, several types of jamming techniques were implemented, and different types of GPS receivers were tested as targets for the spoofed signal in different scenarios. Based on our previous work regarding jamming [20] and spoofing [21], and considering, in addition, the problem of determining the location of the target UAV, the goal of this paper is to design and implement a complete mobile system that can, in a controlled way, land the invading UAV in a geographical point determined by the user, thus, neutralizing the potential threat. Differently from other approaches, the system presented in this paper adopts a simpler philosophy being directed to a compact and localized/independent implementation. The main differences between the proposed system and other existing solutions can be summarized as follows:The system presented in this paper does not depend on third-party systems to determine the location of the UAV.The system presented in this paper does not depend on the detection of specific communication protocols and does not need to carry out any identification of the protocol type used between the UAV and the controller (no sniffing of packages). The system offers a more generalized applicability, since it works for UAVs that use conventional communication schemes and protocols, as well as variations of these.The proposed scheme does not replicate remote control commands to take over the UAV control. The system acquires indirect control of the UAV by deceiving the radionavigation system and not through signals transmitted by the remote control.

Furthermore, the main contributions of this paper can be summarized as follows:We propose and describe a low-cost portable system relying on SDR platforms, which can be manipulated simply by a user, and which is capable of detecting and neutralizing unauthorized drones using electromagnetic waves.We propose and describe the implementation of a spoofing scheme incorporated into the anti-UAV system that consists of first jamming the GPS signal, followed by the transmission of false signals that allow the acquisition and indirect control of the unauthorized drone, enabling the user to land it in a pre-defined location.We propose a simple jamming scheme incorporated into the anti-UAV system that is activated in parallel with the spoofing functionality and that is directed to the control signals so that the vehicle control cannot be accomplished in manual mode.

The remainder of the paper is organized as follows:

Section 2 introduces the global architecture of the system, where the different blocks that constitute the system are presented. Section 3 describes the different jamming techniques considered for the systems, whereas the spoofing technology is presented. Section 4 presents the overall prototype, the radiation diagram of the antenna developed for the desired frequencies and the battery sizing. Interference and spoofing tests in a real scenario are described in Section 5, followed by the conclusions in Section 6. Finally, in Section 7 future work is reported.

## 2. Overall System Architecture

The aim of the system described in this paper is to prevent the intrusion of unauthorized drones, commercial or customized, which includes, for example, land, aerial and water drones operating in restricted areas such as airports, military areas, condominiums, occasional public events or other areas where public safety needs to be guaranteed. Through a touchscreen, the user of the system visualizes a map of the area where he is located, allowing him to select the intended landing spot for an eventual intruder drone. Once an invading drone is detected, the user just needs to point the system at the drone and press the trigger. The system will guide the invading drone to the previously selected site and force it to land. In order to be easily and quickly adapted to the areas to be protected, the system was designed so as to be portable. The modular system incorporates efficient technologies and materials, based on its ease of transportation and handling by defense forces (military and public security). Figure 1 represents the main modules that make up the system.

The user module is composed of controls/commands and a user interface and is, thus, responsible for all user inputs. The battery module is responsible for powering the entire system. The processor module consists of the microprocessor, responsible for all system processing. The sensor module is composed of the microcontroller, which is directly connected to the microprocessor and is responsible for managing and collecting data from the whole set of sensors. The transmission module consists of the electronic equipment required for the transmission of radio signals and the directional antennas. The security module protects the system by resorting to authentication through a biometric sensor, which allows operation only to previously authorized users. Several accessories (e.g., telescopic sight, laser optics, bipod, front grip and sling) are incorporated into the system in order to help improve the user experience and, consequently, contribute to the improvement of performance when operating the system for the neutralization of target UAVs.

As previously stated, the main functionality of the system is the ability to divert the invading drone to a zone determined to be safe. In order to perform this task, the system needs to acquire the location of the target drone. To determine the location of the invading drone, a set of sensors are attached to the system such as: distance meter, accelerometer, location sensor and magnetometer.

In terms of security, to prevent the misuse of the equipment by unauthorized individuals, the system allocates a biometric sensor, as previously stated. Furthermore, through mobile network communications, the system sends a periodic message (time configurable) with the information of the location of the system to a supervising entity. Complementarily, whenever someone initializes the system or initiates the transmission, a message is also sent with the respective username.

This system minimizes interference in third party systems through the use of highly directive antennas.

In summary, the operation of the system can be divided into five stages, as shown in Figure 2.

The first is the authentication step which is needed for a previously registered user to authenticate himself and to initialize the system, using the security module. The second step is the user input required for defining the zone where the invading drone will land. This is accomplished through the user module.

There are two types of jamming implemented in the system: jamming aimed at the communication link between the drone and the remote controller (typically 2.4 GHz) and jamming aimed at the GPS signals received by the drone (around 1.5 GHz). Spoofing of GPS signals is only possible after jamming GPS signals. Therefore, the third step, which corresponds to jamming, comprises two different phases. In the first phase, jamming is applied to the communication between the drone and the remote controller and to the GPS signals. The second phase occurs in parallel to the spoofing steps and corresponds to jamming the control signal only.

The fourth step consists of identifying the location of the invasive drone. The system determines the location of the system using a set of sensors that are managed using the sensor module. The fifth step is one of the main functions of the system and is the responsibility of the Tx module. In this case, this module starts by emitting spoofing signals targeted at the UAV, enabling the system to gain control over the drone. Finally, the sixth step comprises the generation of activity logs (date, time, signals issued, operator and location) which are sent to the supervisory entity.

## 3. System Functionalities

### 3.1. Radio Frequency Interference

Most drones typically operate in one of two modes: the manual piloting mode (controlling the drone with the remote control) and the autonomous mode (previously plotting a route through geographic location coordinates). In both cases, the UAV relies on radio signals. In the case of the UAV being controlled by remote control, radio communications typically at 2.4 GHz or 5.8 GHz are used. When flying autonomously, they resort to GNSS signals. The implemented system can emit efficient signals to block these two types of operation. Since it is difficult to know which flight mode the unauthorized drone is in, the proposed anti-UAV mobile system transmits jamming signals on all frequencies of remote control and GNSS signals.

In [22], we conducted a study to understand which was the most effective approach for jamming the GPS signal. Five different techniques of jamming the GPS signal were considered: barrage jamming; tone jamming; sweep jamming; successive pulses jamming; and protocol-aware jamming. The different techniques were implemented and evaluated using low-cost SDR platforms and the GNU radio software development toolkit. The best-performing jammer was protocol-aware jamming (Figure 3), which uses an architecture similar to that used by a transmitter of GPS signals. Using this approach, the interfering signal mixes more effectively with the main signal since they exhibit identical spectral behavior, requiring a lower transmitter power to successfully disable the correct reception of the GPS signal.

In the case of jamming against remote control frequencies, we employ the technique of barrage jamming, because we have no knowledge of the protocol to be used in the communication. The barrage jamming is the simplest form of interference and is generally defined as a jammer that transmits noise-like energy throughout the portion of the spectrum occupied by the target, as shown in Figure 4. It essentially increases the noise level in the receiver, making it difficult to operate the communication system.

### 3.2. Spoofing

Spoofing, in general, is a fraudulent or malicious practice in which communication is sent from an unknown source, disguised as a source known to the receiver. The use of spoofing is more common in mechanisms and communication networks that do not have a high level of security. In the civil GPS signal, there is not any type of encryption or authentication to protect or to prove that the signal comes from a reliable source or the non-occurrence of repudiation of the signal. Thus, to accomplish spoofing and deceive a GPS receiver, one can simulate GPS signals as if they are coming from real satellites.

The spoofing developed in this project is based on our previous study [21], which considered an open hardware electronic prototype platform, sensors, an SDR module and a system on chip (SoC) as the central processor of the system. The adopted approach used a simple spoofing technique, which could generate and transmit false GPS signals.

The indirect acquisition of the drone control is carried out through the use of the ephemeris files, provided that they allow the simulation of a set of satellites, thus, forming a hypothetical constellation, which subsequently generates baseband signal data streams that are transmitted in the frequency range of the satellite location system. The system was designed with two spoofing modes of operation, one that worked by taking into account only the drone’s location, and another, which took into account not only the location but also the direction and speed. Using these parameters, it is possible to implement and apply the spoofing more smoothly and less noticeably to the target. In [21], the spoofing functionalities were studied and tested on various devices, such as different GPS receivers: smartphones, u-blox M8 GNSS Evaluation Kit, and u-blox MAX-7Q. In this paper, we gave more focus to testing the spoofing functionality integrated into the complete anti-UAV system and targeted at the GNSS receivers in UAVs.

In our implementation, a spoofing trajectory can be specified in either a CSV file, which contains the Earth-centered, Earth-fixed (ECEF) user positions, or an NMEA GGA stream. To illustrate, in the following, we show the example of assigning a static location directly from the command line. First, the user specifies the GPS satellite constellation through a GPS broadcast ephemeris file. The daily GPS broadcast ephemeris file (brdc) is a merge of the individual site navigation files into one (the archive for the daily file can be downloaded from https://cddis.nasa.gov/archive/gnss/data/daily/ accessed on 10 January 2022). These files are then used to generate the simulated pseudorange and Doppler for the GPS satellites in view. This simulated range data is then used to generate the digitized I/Q samples for the GPS signal. As an example of a static mode transmission via the command line we can execute



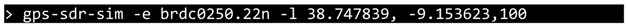



In this command line, option -e <gps_nav> denotes RINEX navigation file for GPS ephemerides (required), brdc0250.22n is the filename and -l <location> is used for providing latitude, longitude and height coordinates (static mode). This command creates a.bin file corresponding to the simulated GPS signal file, named “gpssim.bin”, which can be loaded into the bladeRF board for playback as exemplified below:



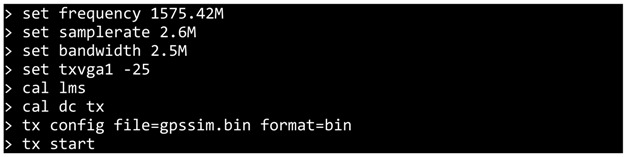



After executing the ‘tx start’ command, BladeRF starts the transmission and will be able to deceive the target receiver.

### 3.3. Localization

There are already several systems that can detect invading drones through a radar. These radars could be the first device to identify the invading drone, as they work autonomously. However, there must be a complementarity/cooperation of both systems (localization and anti-UAV techniques). To determine the exact location of the UAV, the implemented system integrates three sensors and a receiver.

Figure 5 illustrates how the estimation of the position of the intruder drone is carried out. The LIDAR Lite v3 [22] is responsible for measuring the distance to the UAV, which is marked as d) in the image. We can project a right triangle between the system and the drone, as is visible in Figure 5. Through the 3D accelerometer MPU6050 [23], which is used for tilt/pitch measurements, it is possible to determine the angle p). With the angle p) and the distance d); using simple trigonometry, it is possible to determine the distance between the drone and the system in relation to the earth plane. Finally, using the e-compass LSM303D [24], which is responsible for measuring the system orientation, m), we can obtain the direction of the drone. By combining the information obtained from this set of sensors with the known location of the system itself, it is then possible to determine the location of the UAV [21]. Note that the system position is estimated through the GPS signal receiver [25].

### 3.4. System Security

Technology is a double-edged sword, and we must always see both sides. Along with the benefits of each step forward comes the possibility that it may be used for malicious purposes. Some examples of what is possible if there is a misuse of the anti-UAV system is the possibility that a malicious user may be able to divert drones in order to keep them in his possession. Furthermore, it could also be used with the aim of interfering and diverting the route of a self-driving car with potentially dangerous consequences.

For these reasons, it is important to include security measures in a portable anti-UAV system such as the one described in this paper. In the proposed implementation, only previously authorized users can use the system to transmit jamming or spoofing signals. The fingerprint scanner—TTL (GT-521F32), chosen as a biometric sensor due to its small and compact size, was used for system implementation. The fingerprint scanner primarily converts biometric information, i.e., the surface or subsurface of the skin of a finger, into a digital signal, typically a digital image. In practice, this conversion process can never be perfect. The persistent and largely invariant part of the imperfections induced by the fingerprint scanner in this process is called the scanner pattern [26]. By switching on the anti-UAV mobile system, the user is required to authenticate himself with a biometric sensor in order to boot up the system [27].

Once the fingerprint details are stored, whenever the fingerprint is scanned the identity can be authenticated. To build a complete and more accurate fingerprint image, the scanner often asks to take multiple prints from the same finger.

As a means for preventing and enhancing system security through mobile network communication, the system sends a periodic (configurable time) message with the location information of the system to a supervisory entity. In addition, whenever someone initializes the system or initiates the transmission, a message with the respective username is also sent. For the communication with a supervisory entity, later identified, the Arduino SIM900 shield was used, as it allows sending messages using general packet radio service (GPRS) technology. By sending the messages to the supervisor entity, it is possible to create a timeline of system usage. This type of monitoring is necessary to increase the security of system operation, by recording all its activity.

## 4. Prototype System

The objective of this work was to build a final prototype of an anti-UAV system, which should be portable and easily manageable by an operator.

The design of the implemented prototype is shown in Figure 6. The main hardware supports the implementation of three distinct modules, which were introduced in Figure 1. The processor module, which is the brain of the entire system being responsible for all interconnecting sensors/actuators, is implemented using a raspberry pi 3 board. A microcontroller which is part of two different modules, the sensor module and the security module, is implemented using an Arduino uno (AtMega329p microcontroller). It has the functionality of collecting and processing all the information coming from the sensors. The Tx module is responsible for transmitting jamming and spoofing signals and is implemented on a software defined radio platform, namely, the BladeRF x40.

To create and transmit the jamming signals, the GNU radio toolbox was used [20], whereas the spoofing signals are generated resorting to the simulator GPS-SDR-SIM available on GitHub [21].

### 4.1. Yagi Antenna Sizing

In order to minimize interference with GPS receivers in the vicinity of the target UAV, there was the need for implementing a directive antenna. With a smaller transmission beam, it is possible to direct the signal to the target and, thereby, to increase the power of the signal in that direction while reducing unwanted interference. The Yagi antenna is a high-power antenna that can be used to transmit signals over relatively long distances, as well as to pick up weak signals. In wireless networks, Yagi antennas are the ones that offer the longest range, but they can only cover a small area where they are aimed. For the Yagi antenna design, a Yagi Calculator—VK5DJ simulator developed by John Drew [28] was used. Yagi Calculator—VK5DJ is a program whose purpose is to produce dimensions for a DL6WU style Yagi antenna. It is designed for frequencies between 144 MHz and 2.4 GHz. The Yagi DL6WU model is considered to be easy to build and with very positive results. For a frequency of 1575.42 MHz and with simulator recommendation to use a minimum of eight directors, the results of the antenna dimensions are expressed in Figure 7a. The simulator also presents in its results the dimensions of the folded dipole in Figure 7b.

The simulator also calculates the antenna gain, which was 14.8 dB with approximately 29° beamwidth for both planes (vertical and horizontal).

The final result of the handmade Yagi antenna, respecting the dimensions provided by the simulator, is shown in Figure 8.

As shown in Figure 9, the radiation diagrams of the handmade Yagi antenna were obtained for the XY (a) and YZ (b) planes. Spectral power density was measured with the RTL SDR 820T2 with five degrees spacing. The blue line represents the measured values. The red line represents the trendline.

It is important to note that the conditions under which measurements were taken to draw the diagrams were not ideal. For the results to be more assertive, measurements should have been made in an anechoic chamber to avoid reflections that influence the intended results.

The angular distance between half power points is defined as the beamwidth [29]. The beam width in both planes XY, Figure 9a; and YZ, Figure 9b, corresponds to a beam width of about 5 degrees for both cases.

To calculate the antenna gain, it is necessary to use the free space propagation model (Equation (1)), since the radiation diagrams presented refer to the received powers.
(1)$$P_{RX}=P_{TX}{\left(\frac{c}{4\pi df}\right)}^{2}G_{TX}G_{RX}$$

As $P_{RX}$ should be equal in both planes, for the purpose of calculations, an average of the two measured values was taken, resulting in $P_{RX}$ = −21.5 dBW. Using this value, we can write
(2)$$G_{TX}=\frac{P_{RX}}{P_{TX}{\left(\frac{c}{4\pi df}\right)}^{2}G_{RX}}=\frac{10^{\frac{-21.5}{10}}}{10^{\frac{-3}{10}}\times {\left(\frac{3\;\times \;10^{8}}{4\pi \;\times \;1.10\;\times \;1.57542\;\times \;10^{9}}\right)}^{2}\times 1}=74.43$$
(3)$$G_{TX}\left(dB\right)=10\log\left(74.43\right)=18.72\;\mathrm{dB}$$

The signal emitted through GNU radio for measuring the values had a transmission gain of −4dB, so the actual antenna gain is:(4)$$G_{Antena}\left(dBi\right)=G_{TX}\left(dB\right)+G_{TX\;GNU\;Radio}=18.72+\left(-4\right)=14.72\;\mathrm{dBi}$$

Table 1 compares the resulting simulator values with the measured values.

The results obtained through the simulator, compared to the experimental values, were slightly different regarding the half power beam width, which translates into the antenna directivity. In this case, a more directive antenna than expected was obtained. Values may differ due to fabrication imperfections and due to measurement conditions, which were not perfect. Regarding antenna gain, the result was very close to the expected one.

A low-cost amplifier, SPF5889Z was used, as we can see in Figure 10. It is important to note that, while not being the ideal power amplifier, it was sufficiently capable for the outlined objective. According to the specifications, the SPF5189Z possesses a gain of 17.9 dB of at 1000 MHz and 13.8 dB at 1700 MHz.

### 4.2. Battery Sizing

As the objective was to implement a mobile anti-UAV system, it is necessary to perform the sizing of the battery to be adopted.

Considering that the adopted Raspberry Pi 3, Arduino Uno and BladeRF work with a voltage of about 5 V, the power supply module was designed so as to supply the same voltage to all these boards. The BladeRF, which was used for generating the jamming and spoofing signals, according to Nuand’s official specifications, possesses a maximum consumption of 3644 W. To increase the transmitted power, we also adopted a Nuand XB-300 board. The Nuand XB-300 is an amplifier expansion card that greatly increases the range of the BladeRF. On the receive side, the XB-300 featured a low-noise amplifier (LNA) and a combiner for antenna diversity. The transmit side featured a power amplifier (PA), TRX switch, and a highly accurate ADC for measuring output power of the PA. The PA exhibited a typical gain of 20 dB at 2.45 GHz with a bandwidth of 100 MHz (2.4–2.5 GHz). The BladeRF, combined with the XB-300 module with Tx ON and Rx OFF, exhibited a consumption of 8584 W; consumptions available from the official Nuand specifications.

As there is no reference amplifier for jammer signal spoofing and GPS amplification, an average value of 1 A was used as a reference. Total instantaneous current consumption was 3.74 A. In Table 2, we summarize the hardware consumption.

If we assume a hypothetical military scenario where a soldier works an 8 h shift and the battery is intended to be able to power the system about 20% of the shift time (1.6 h), then we can roughly estimate the required capacity as:(5)Capacity=autonomy (hours)×consumption≈ 6 Ah

## 5. Real Environment Tests

Several tests were accomplished in order to evaluate the performance of the complete anti-UAV mobile system under the influence of real GNSS signals. It is important to note that, in these tests, we focused mostly on the evaluation of the overall system against drones operating in automatic flight mode. Still, in a previous work of some of the authors [30], we had already presented several experimental interference tests using the adopted jamming solution, which showed that it was possible to disrupt the UAV communication link.

For the tests presented in this paper, the BladeRF was powered by USB 3.0, with gains of TxVGA1 = −4 dB and TxVGA2 = 25 dB.

For the operation of the system, it is necessary to select the forced landing site, intended for the forced landing of possible invading drones. We can see the selection adopted for the tests in Figure 11.

In following figures, we illustrate the real test scenario, which corresponded to a rural environment, wherein it was possible to visualize the roof of a one-story house, bushes, some ground vegetation and a few trees. The tests were carried out in a fully controlled and safe environment.

The drone used for the real-life tests was a DJI Phantom 3 Standard, illustrated in Figure 12.

Selecting the auto flight mode and with the drone already in flight, whenever the jammer signal transmission was started, the drone’s behavior automatically changed, stopping its flight path and gliding in the same place.

It was observed that the time between starting jammer transmission and drone gliding is almost instantaneous.

If the jammer is started before the drone takes off, the drone will not even perform any action.

All tests were performed safely, away from air traffic and outside prohibited areas.

After the drone halts its operation due to jamming it was then possible to emit spoofing signals so that the drone was deflected and indirectly controlled.

Figure 13 shows the test scenario. The UAV was pre-configured with an autonomous route marked in red in the image. As soon as the operator started transmitting jamming signals, the drone was hovering in the same place (ball marked in red).

The remaining pre-configured route was not completed by the UAV.

The anti-UAV mobile system operator (Figure 14) changed the mode of operation from jamming mode to spoofing mode.

With the operator pointing towards the target UAV, using the telescopic sight, if necessary, the geographical determination of the UAV was initiated.

Through the sensor module, the location of the UAV was determined and, through the processor module, the spoof route to the forced landing site was generated.

We can observe the whole scenario in Figure 15.

When the UAV received the spoofing signals, it assumed its new route and moved to the forced landing site as desired. The system’s performance was positive, and we managed to divert the drone at the four cardinal points. The UAV behavior was as expected with the theory; the spoofing behaved as shown previously in [21] regarding several well-known GPS receivers.

## 6. Conclusions

In this paper, we described an anti-UAV mobile system that was developed in order to protect areas from unauthorized drones. The system incorporated localization with jamming and spoofing techniques, which were used in a coordinated approach. The system was able to estimate the position of the invading vehicle and could then apply jamming to neutralize the GPS signal, followed by spoofing to gain control of the vehicle. In addition, it also supported jamming directed to the control signal, thus, making manual control unfeasible.

Through carefully conducted real-life tests using the implemented prototype, we observed the effectiveness of the proposed solution, which was capable of blocking the autonomous flight of a UAV by jamming its GPS receiver, followed by the successful acquisition of the vehicle control through the transmission of spoofing signals, which allowed its forced landing on a predefined site.

The proposed system can be used with the aim of helping to solve problems related to security and privacy. As it is a mobile system, it can be easily adapted to different geographical areas and different scenarios. As possible applications, it can be used for protection against possible terrorist attacks at political speeches, festivals, sporting events or others, where terrorists may resort to bombs attached to drones. It can be used to reduce the trafficking of illegal substances and objects into prisons, where criminals use drones to transport them by air. It can also help to solve situations of negligent drone pilots who fly these devices to capture images of forest fires, obstructing aerial firefighting operations. There are many other cases and situations where this system can act as a non-destructive approach capable of forcing an invading drone to land.

## 7. Future Work

As future work, besides extending the maximum operational range using better amplifiers and antennas with higher directivity, the authors intend to extend the system for fixed position surveillance applications, where it can be deployed for continuous monitoring of a predefined area. In this case, beamforming technologies can be applied, thus, enabling it to defend against an intrusion of multiple unauthorized drones. In fact, in this specific type of scenario, with a multiple drone attack, the system presented in this paper will have difficulties in being able to provide an effective response. In this situation, an individual system and an operator would be needed for each invading drone, which makes the solution less appealing. Therefore, for this type of scenario, a fixed system incorporating beamforming technologies can constitute a better approach.

## Figures and Tables

**Figure 1 sensors-22-01487-f001:**
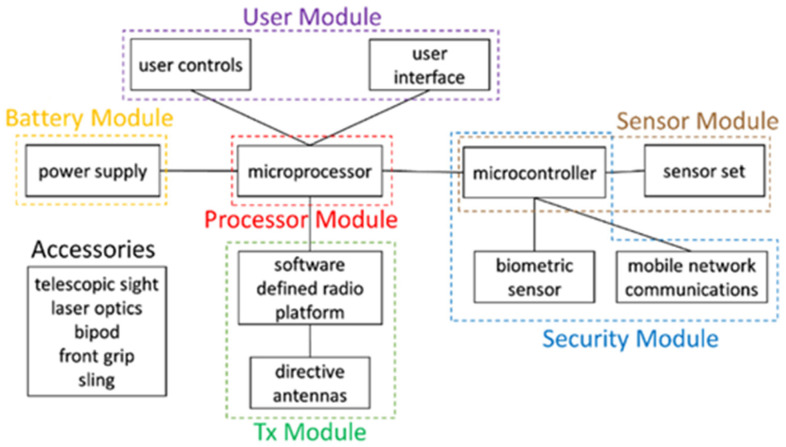
System architecture.

**Figure 2 sensors-22-01487-f002:**
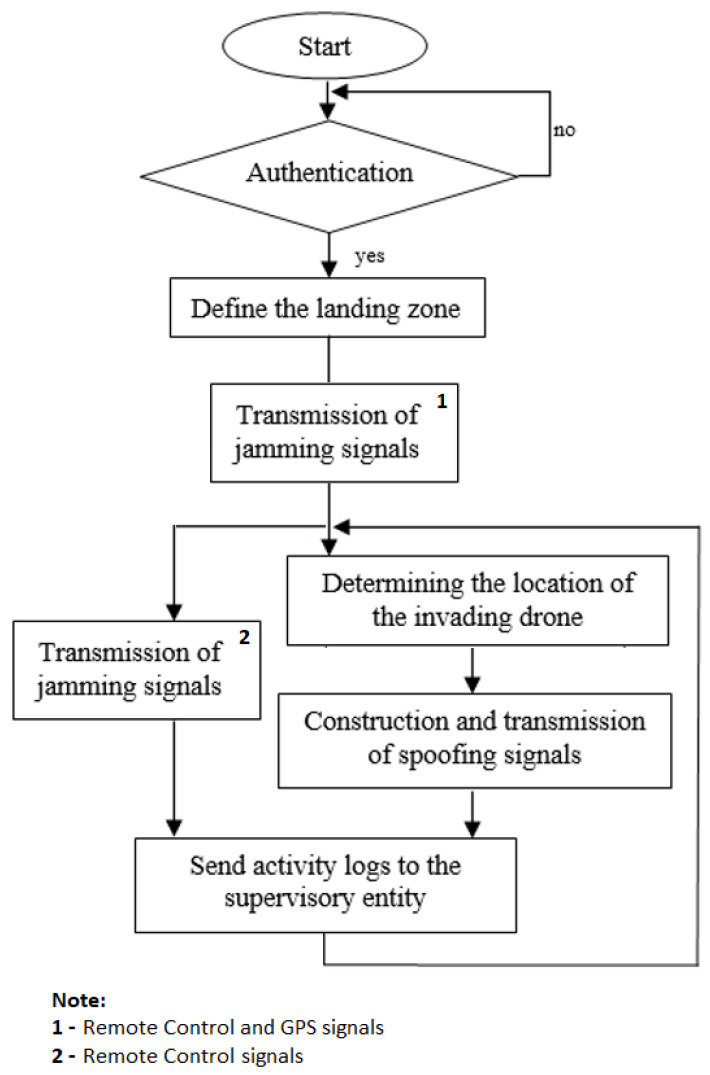
System flowchart.

**Figure 3 sensors-22-01487-f003:**
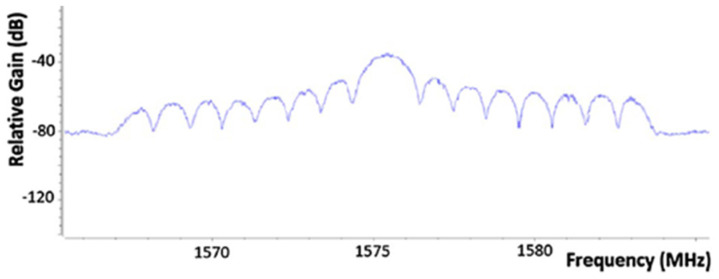
Protocol-Aware Jamming Spectrum.

**Figure 4 sensors-22-01487-f004:**
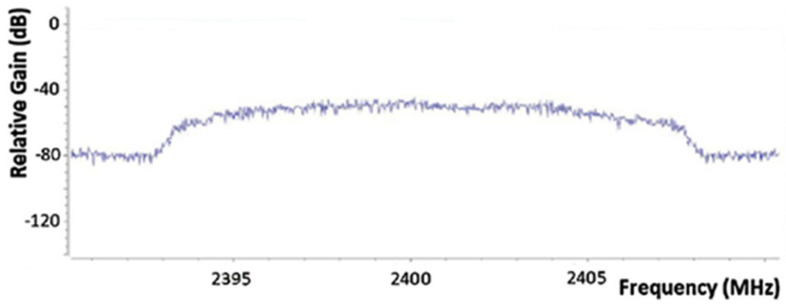
Barrage Jamming Spectrum.

**Figure 5 sensors-22-01487-f005:**
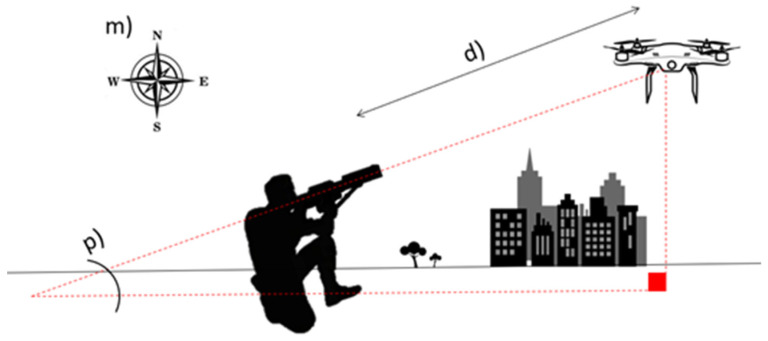
UAV localization method.

**Figure 6 sensors-22-01487-f006:**
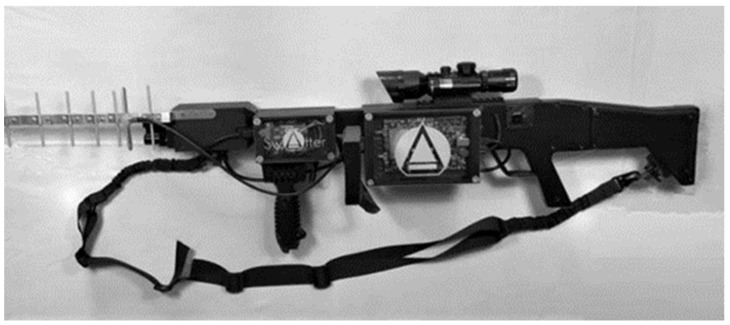
System Anti-UAVs.

**Figure 7 sensors-22-01487-f007:**
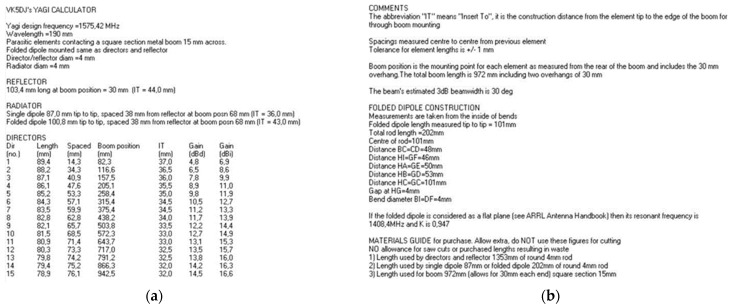
This are the results of a VK5DJ’s Yagi Calculator. (**a**) Yagi Antenna Sizing Results; (**b**) Results of folded dipole dimensions.

**Figure 8 sensors-22-01487-f008:**
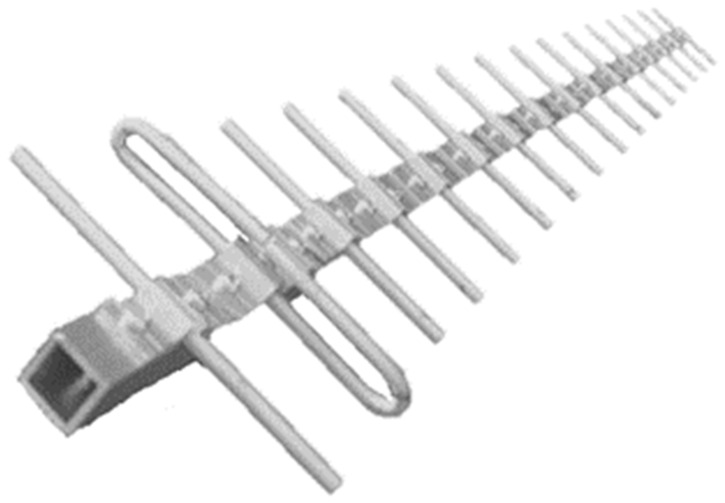
Handmade Yagi Antenna.

**Figure 9 sensors-22-01487-f009:**
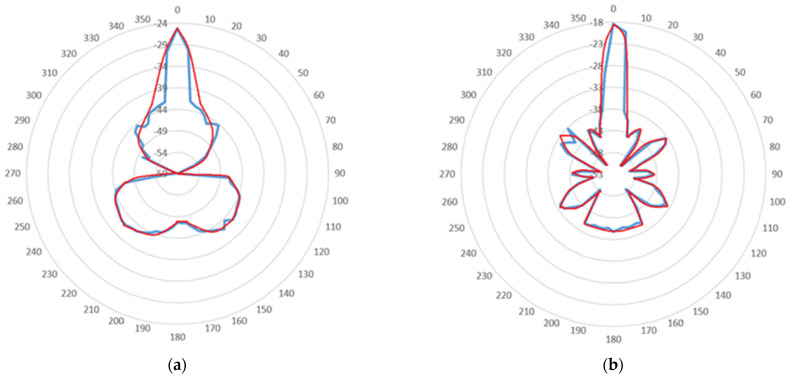
Radiation diagrams of the handmade Yagi antenna (**a**) XY plane radiation diagram; (**b**) YZ plane radiation diagram.

**Figure 10 sensors-22-01487-f010:**
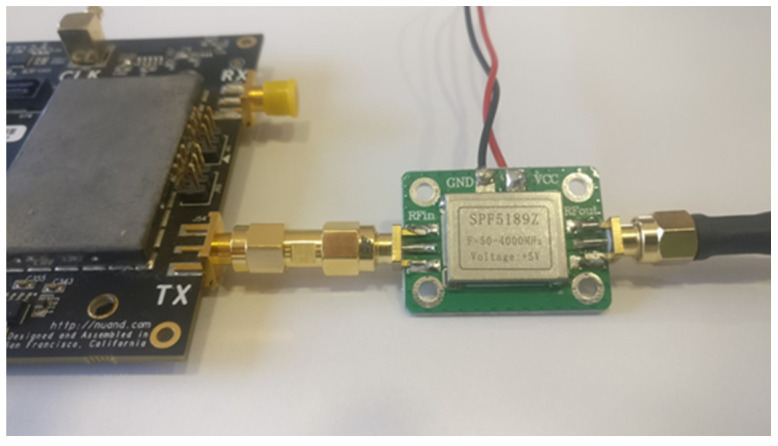
Power amplifier SPF5189Z.

**Figure 11 sensors-22-01487-f011:**
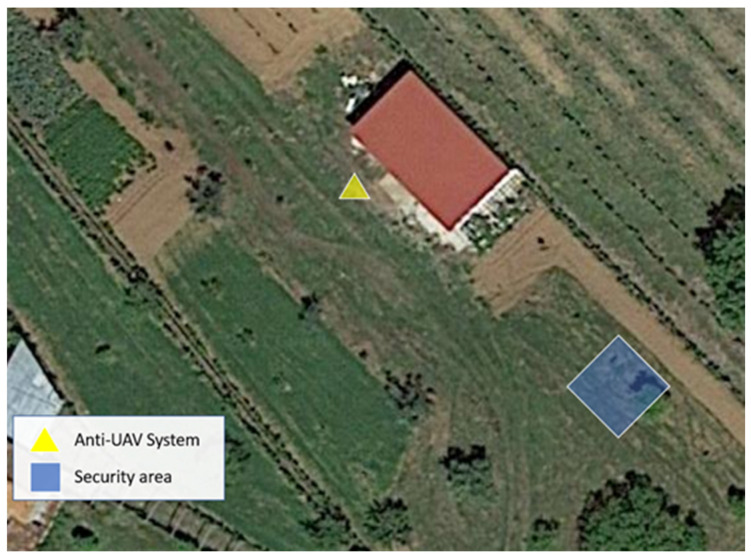
Real test scenario with selected forced landing site.

**Figure 12 sensors-22-01487-f012:**
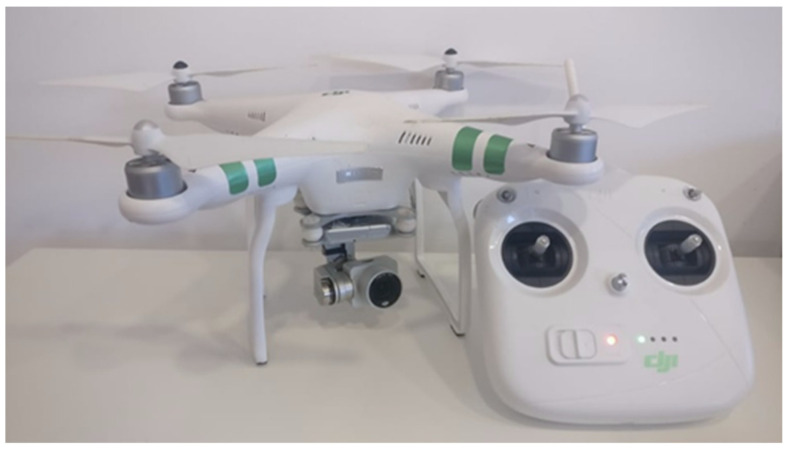
DJI Phantom 3 Standard.

**Figure 13 sensors-22-01487-f013:**
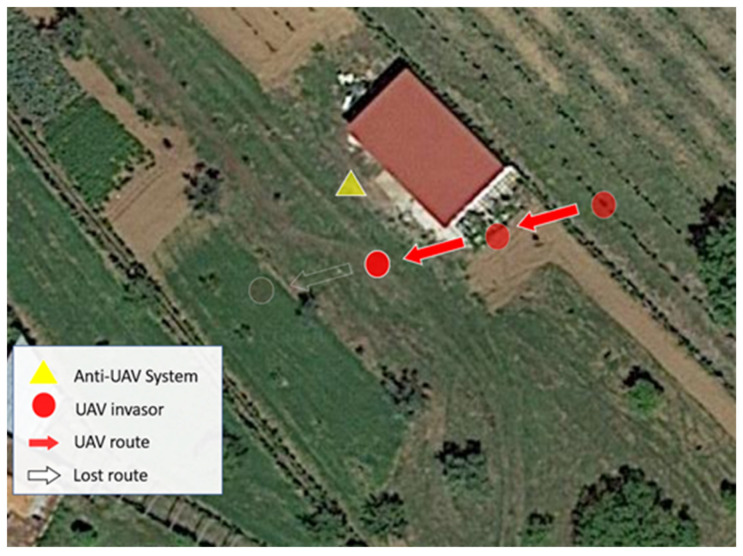
Real test scenario using jamming signals.

**Figure 14 sensors-22-01487-f014:**
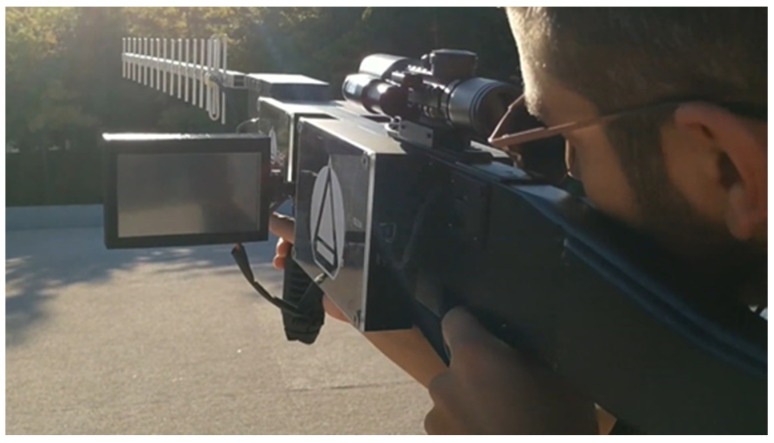
Anti-UAV mobile system operator.

**Figure 15 sensors-22-01487-f015:**
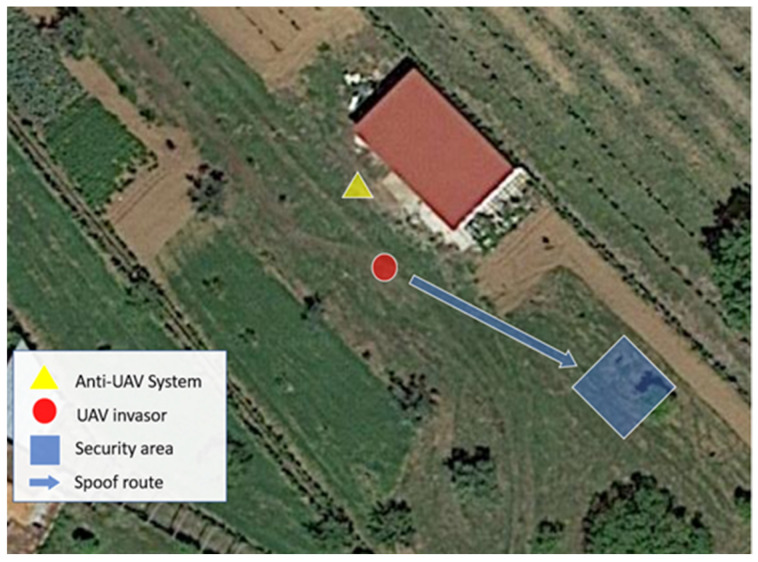
Real test scenario using spoofing signals.

**Table 1 sensors-22-01487-t001:** Spectral power density of different jammers.

	Half Power Beam Width	Antenna Gain
Simulator Values	29 degrees	14.8 dBi
Real Values	5 degrees	14.72 dBi

**Table 2 sensors-22-01487-t002:** Hardware current consumption.

Hardware	Ampere
BladeRF x40	0.73
BladeRF x40 + xb300	1.72
Arduino Uno	0.05
Raspberry Pi 3	0.24
Power Amplifier (SPF5189Z) (≈1.5 GHz)	1
Total	3.74

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
