# Peer review of "A Software Defined Radio Based Anti-UAV Mobile System with Jamming and Spoofing Capabilities"

_sensors, 2022, doi:10.3390/s22041487_

Round 1
Reviewer 1 Report
Line 122-125: What are the accessories used for improving user experience? Line 141: System Flowchart. In the flowchart, transmission of jamming and spoofing signals are shown as if they are performed in parallel. However, in the description of the tests (Line 449-450), it is stated that spoofing is possible only after jamming. Therefore, the flowchart should be corrected accordingly. See also comment about Line 464-467. Line 146: How is the landing zone defined in the system? By entering GPS coordinates? Please specify. Line 153-155: Why is there a need to send the massages to the supervisory entity in real-time? Please identify. Couldn't the logs be recorded and then downloaded offline? Line 218: What is meant by "both systems"? Radar and Anti-UAV? Please identify. Line 236-241: Need to identify why there's a need for user authentication. Provide case scenarios or historical events where such an anti-UAV system can be or has been used for malicious purposes. Line 249-259: No need to give the details on how the biometric system works, including the figure 6. It is not relevant to the topic of the paper. Line 292: The use of GPR-SDR-SIM tool (from GitHub) with the spoofing method identified by the authors need to clarified. Line 331-333: The measurement of spectral density was performed with 5 degree spacing, and then half-power beamwidth was also found to be 5 degrees. The researchers should use a smaller degree spacing (like 1 degree) to measure such a beamwidth. The issue is about precision and error in the measurement. In the measurement presented, the error rate is in the range of the measurement. Line 405-407: The gain of the power amplifier of XB-300 is given for 2.45 GHz. But this is not the operating frequency of the system. The gain value for the operating frequency (around 1.5 GHz) should be given. Line 424: It is stated that several tests were performed but minimal numeric data is provided, e.g., distance to UAV, tilt angle, etc. to identify the maximum range of the system. Line 441: Tests are performed only in flight-mode, no result was presented when the UAV is manually operated. Line 464-467: The localization of the drone (Step 3) is performed after jamming (Step 4). This is not consistent with the system flowchart given in Figure 2. Line 468-473: The information on power amplifier should be given in Section 4, not in Results. Line 468-470: In the prototype description, we've learned about the use of XB-300, how is this new power amplifier connected to the system? What is its contribution to the gain of the system? Minor/Typos: Line 198: "as if they came from' --> "as if they are coming from" Line 218: "to determinate" --> "to determine" Line 216-232: It can be shortened into one paragraph removing repetitions. Line 262: A new paragraph start with the sentence "As a means for preventing ...." Line 267: "Arduino SIM 900 shiel" --> "Arduino SIM 900 shield" Line 434: Figure 11 caption to be correctedAuthor Response
Thanks for your review. I send a word file with a point-by-point response to the reviewer’s comments

Reviewer 2 Report
- Some sentences are too redundant and unclear, such as lines 102 to 105.
- The ‘Software Defined Radio’ in the title is difficult to be reflected in the paper. You can explain this concept in more detail.
- Software-defined networks and related topics should be discussed. A Novel Prediction-Based Temporal Graph Routing Algorithm for Software-Defined Vehicular Networks” IEEE Transactions on Intelligent Transportation Systems (T-ITS), 2021. “Intelligent Content Caching Strategy in Autonomous Driving Towards 6G,” IEEE Transactions on Intelligent Transportation Systems (T-ITS), 2021. Some other research on jamming UAV can be introduced. At present, there is little introduction to the related work, mainly to introduce the research of related jamming system.
- The experimental schematic diagram in the article can be more detailed. Although the experimental results can be understood from the diagram, the result diagram is too simple and unclear and not very convincing. Especially Fig. 11, 13 and 16.
- You can consider listing the contributions of the whole system, so that readers can better understand the advantages of this new design system.
- The introduction of some technology is too simple and abstract, and can further receive the working mechanism of each module.
Author Response
Thanks for your review. I send a word file with a point-by-point response to the reviewer’s comments

Reviewer 3 Report
The article appears to be technically sound; nonetheless, the authors must address the following shortcomings by improving the manuscript or clarifying their response:
- The authors should clearly state that how their system differs from existing systems. Because the authors provided relatively limited literature, readers may have difficulties understanding the research work key contributions. If feasible, segregate the contribution portion to make it easier to understand.
- The authors need to redraw figure 3, figure 4 and figure 8 in a higher resolution.
- Conclusions section is poorly written.
- Authors should include future work, which may help other researchers to work in this interesting field.
- Very weak references are provided by the authors. Authors need to improve the references part. Add more relevant references from some good journals if available.
- Authors need to correct typographical, punctuation, and grammar errors throughout the paper.
Author Response

(The authors gave the same response as above.)

Round 2
Reviewer 1 Report
Thanks for incorporating comments and improving the paper.Reviewer 2 Report
No further comments.